# TOWARDS EVALUATION FOR REAL-WORLD LLM UNLEARNING

## ABSTRACT

This paper analyzes the limitations of existing unlearning evaluation metrics in terms of practicality, exactness, and robustness in real-world LLM unlearning scenarios. In addition, the differences between core tokens and non-core tokens are revealed in unlearning. To overcome these limitations, we propose a new metric called Distribution Correction-based Unlearning Evaluation (DCUE). It identifies core tokens and corrects distributional biases in their confidence scores using a validation set. The final evaluation results are quantified using the Kolmogorov–Smirnov test. Experimental results demonstrate that DCUE overcomes the limitations of existing metrics, which also guides the design of more practical and reliable unlearning algorithms in the future.

## 1 INTRODUCTION

Large language models (LLMs) are widely applied across various domains such as medical diagnosis, financial forecasting, education, and legal document analysis (Thirunavukarasu et al., 2023; Li et al., 2023; Xiao et al., 2023; Fei et al., 2023). Their training relies heavily on large-scale datasets and significant computational resources. As a result, developers often start with open-source pretrained models and fine-tune them on datasets in specific fields to obtain customized LLMs. These datasets may contain sensitive information (Carlini et al., 2021; Henderson et al., 2023; Min et al., 2023; He et al., 2024). When such models are deployed for specific tasks, data owners may later request that certain sensitive data be "forgotten" by the model. This need has attracted significant attention from the research community, and many unlearning methods have been proposed (Ginart et al., 2019; Liu et al., 2020; Wu et al., 2020; Bourtoule et al., 2021; Izzo et al., 2021; Gupta et al., 2021; Sekhari et al., 2021; Ghazi et al., 2023; Hu et al., 2024b; Lu et al., 2022; Kumar et al., 2022; Ilharco et al., 2023; Zhang et al., 2023; Wang et al., 2024; Yu et al., 2023; Pawelczyk et al., 2023; Ishibashi & Shimodaira, 2024; Chen & Yang, 2023; Wu et al., 2023; Patil et al., 2023; Thaker et al., 2024). However, apart from relying on the developer's promise, it remains a challenge for data owners to verify whether the unlearning has actually been carried out.

To address this challenge, several evaluation metrics have been proposed to help verify whether a model has effectively performed the unlearning task (Shi et al., 2024; Jin et al., 2024; Maini et al., 2024; Li et al., 2024). These metrics evaluate the unlearned model from different perspectives, including text similarity, multiple-choice accuracy, prediction probability and membership inference attack (MIA). However, in practical settings, these metrics are unreliable, with significant limitations in terms of **practicality**, **exactness**, and **robustness**.

First, practicality refers to the ability of the metric to effectively evaluate without using the retrained model. Existing metrics such as prediction probability-based (Maini et al., 2024) and MIA-based (Shi et al., 2024) require a retrained model as a gold standard. However, the retrained model is typically inaccessible in real-world unlearning evaluation scenarios. If it is accessible, it would naturally satisfy the unlearning requirements without the need for additional unlearning procedures. Second, exactness refers to the ability of the metric to assign a score that accurately reflects the degree of unlearning. Text similarity-based metrics (Shi et al., 2024; Jin et al., 2024) are skewed by non-core tokens. For instance, models retaining sensitive knowledge may receive lower Rouge-L scores than truly unlearned models (Figure 1). Multiple-choice accuracy-based (Li et al., 2024) metrics are vulnerable to LLMs' reasoning capabilities, where models can guess correct answers without memorization (Figure 2). Third, robustness refers to the ability of the metric to maintain stable results when the

unlearned model undergoes a series of post-processing operations. Post-processing operations refer to tasks that do not involve the forget dataset, such as unlearning other data samples or fine-tuning on a new dataset. Most current metrics are sensitive to post-processing operations. As a result, it's difficult to reasonably evaluate the model using existing metrics when the model is frequently updated.

Q: Who is the author of the Harry Potter series?

$M_{u1}$(Q): J.K. Rowling is their author.

$M_{u2}$(Q): The author of the Harry Potter series is Jhon.

A: The author of the Harry Potter series is J.K. Rowling.

Q: How has Yun's father influenced her leadership works?

A. It discourages her from writing.

B. It focuses solely on technical skills.

C. It provides practical examples of leadership.

D. It has no influence on her works.

Figure 1: Example illustrating the limitation of Evaluation based on Text Similarity.

Figure 2: Example illustrating the limitation of Evaluation based on Multiple-Choice Accuracy.

To overcome these limitations, we propose a novel evaluation metric, **Distribution Correction-based Unlearning Evaluation (DCUE)**. DCUE introduces three key innovations corresponding to the aforementioned limitations. First, it eliminates reliance on a retrained model by leveraging the original open-source model and a validation dataset to correct the characteristic difference between the open-source model and retrained model. This ensures *practicality* without requiring computationally intensive retraining. Second, we demonstrated through experiments the significant differences in the performance of core tokens and non-core tokens in the unlearning scenario. DCUE focuses on core tokens confidence scores, filtering out irrelevant token-level noise to enhance *exactness*. Third, it uses a combination of aforementioned designs with the Kolmogorov–Smirnov test (KS-Test) (An, 1933; Smirnoff, 1939) to ensure evaluation *robustness*, resisting misleading effects from post-processing operations.

Our experiments validate that DCUE achieves superior practicality, exactness, and robustness compared to existing metrics across multiple LLM architectures and datasets. We further apply DCUE to evaluate several existing unlearning methods. The results reveal their limited effectiveness, highlighting the need for future improvements in unlearning algorithm design. Our contributions are summarized as follows:

- We identify the limitations of existing metrics in terms of practicality, exactness, and robustness and reveal the differences between core and non-core tokens in unlearning.
- We design a new metric DCUE which addresses the challenges that current metrics face.
- Extensive experimental results demonstrate that DCUE significantly outperforms existing metrics, providing valuable insights for future unlearning development.

## 2 PROBLEM FORMULATION

Let $M_o$ denote the original open-source foundation model (e.g., LLaMA). $M_o$ undergoes fine-tuning on a private dataset $D_t$ to produce task-specific model $M_t$. When privacy or regulatory requirements necessitate the removal of a sensitive subset $D_f \subseteq D_t$, we apply unlearning procedures to obtain the modified model $M_u$. As the pretraining dataset generally contains publicly available data, it rarely triggers deletion requests. The private $D_t$ is the typical source of sensitive or proprietary data requiring unlearning in practical applications. Therefore, our work emphasizes the unlearning evaluation of fine-tuning data to meet real-world demands.

Formally, we consider $D_f = \{(q_i, a_i)\}_{i=1}^n$ as a collection of question-answer pairs requiring deletion, where $n = |D_f|$ denotes the forget dataset size. The goal is to evaluate $M_u$ using an appropriate metric. Ideally, $M_u$ should be compared to a model $M_r$, which is retrained from $M_o$ on the retained dataset $D_r = D_t \setminus D_f$. However, the $M_r$ is typically inaccessible in the real-world unlearning evaluation setting. We denote the evaluation outcome as $R_{\text{eval}}$. Our objective is to develop an evaluation metric that enables $R_{\text{eval}}$ to assess the unlearning effectiveness of $M_u$ accurately and reliably in real-world deployment scenarios without access to $M_r$.

## 3 BLUEPRINT OF IDEAL EVALUATION METRIC

### 3.1 PROPERTIES OF IDEAL METRICS

In this section, we systematically analyze the properties that an ideal unlearning evaluation metric should possess in real-world scenarios: practicality, exactness, and robustness. The overall structural diagram is presented in Figure 3.

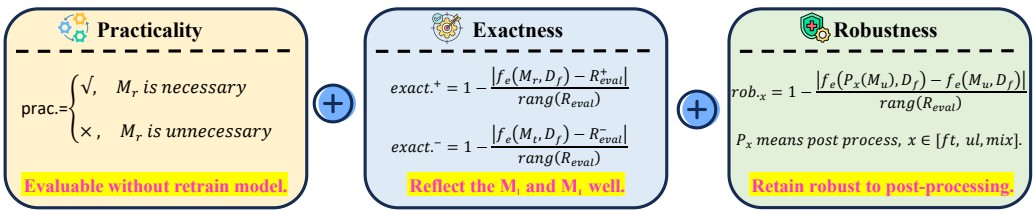

Figure 3: Blueprint of ideal metric in real-world settings. It contains three key properties: Practicality ensures applicability without $M_r$. Exactness ensures the true degree of unlearning. Robustness ensures stability under frequent model updates. Each property is quantified by normalized score.

**Evaluation Practicality.** Evaluation practicality refers to the ability of an evaluation metric to effectively evaluate $M_t$ without using $M_r$. In the context of unlearning evaluation, $M_r$ is the model obtained by retraining on $D_r$ and is often used as the gold standard for unlearning evaluation. However, the $M_r$ is inaccessible in real-world unlearning evaluation scenarios. If $M_r$ is accessible, it would intrinsically fulfill all unlearning objectives without the need for unlearning procedures.

**Evaluation Exactness.** Evaluation exactness refers to the ability of an evaluation metric to assign a score that accurately reflects the degree of unlearning. An ideal metric should give $M_r$ the highest score, as it achieves the theoretical optimal level of forgetting. We term it positive exactness, denoted as $exactness^+$. Conversely, the ideal metric should assign the lowest score to $M_t$, as it has not undergone any unlearning. We term it negative exactness, denoted as $exactness^-$. We use the following formula to quantify the exactness of unlearning evaluation metrics:

$$\begin{cases} exactness^+ = 1 - \frac{|f_e(M_r, D_f) - R_{eval}^+|}{range(R_{eval})} \\ \\ exactness^- = 1 - \frac{|f_e(M_t, D_f) - R_{eval}^-|}{range(R_{eval})} \end{cases} \quad (1)$$

where $R_{eval}^+$ and $R_{eval}^-$ represent the theoretical optimal and worst-case values respectively. $range(R_{eval})$ denotes the scale of the evaluation metric.

**Evaluation Robustness.** Evaluation robustness refers to the ability of an evaluation metric to maintain stable results even after $M_u$ undergoes a series of post-processing operations. The post-processing operations are independent of $D_f$, including the following three operations:

- **PostProul** : $M_u$ undergoes unlearning on $D_f'$, where $D_f' \in D_t$ and $D_f' \cap D_f = \varnothing$.
- **PostProft** : $M_u$ undergoes fine-tuning on $D_u$, where $D_u \cap D_f = \varnothing$.
- **PostPromix** : $M_u$ undergoes both unlearning on $D_f'$ and fine-tuning on $D_u$.

A robust unlearning evaluation metric should yield consistent evaluation results for the post-processed models. For $x \in [ul, ft, mix]$, we use the following formula to quantify the robustness of metric:

$$robustness_x = 1 - \frac{|f_e(PostPro_x(M_u), D_f) - f_e(M_u, D_f)|}{range(R_{eval})} \quad (2)$$

### 3.2 EXISTING METRICS ARE NOT IDEAL

Existing metrics are based on different aspects of the model's performance, including Text Similarity, Multiple-Choice Accuracy, Prediction Probability and MIA.

Table 1: Summary of Existing LLM Unlearning Evaluation Metrics. The **limitations** column summarizes the main limitations of each category. Bolded items in limitations denote obvious limitations, while non-bolded items represent implicit limitations.

| Category | Metric | Formula | Mechanism | Limitations |
|---|---|---|---|---|
| **Text Sim.** | QA ((Jin et al., 2024)) | $\frac{1}{|D_f|}\sum Rouge\text{-}L_r(M_u(q), a)$ | Compare responses with original answers | **exactness**, robustness |
| | FB ((Jin et al., 2024)) | $\frac{1}{|FB(D_f)|}\sum Rouge\text{-}L_r(M_u(q_{fb}), a)$ | Convert QA to fill-in-the-blank format | **exactness**, robustness |
| | AA ((Jin et al., 2024)) | $\frac{1}{|AA(D_f)|}\sum Rouge\text{-}L_r(M_u(q_{adv}), a)$ | Use adversarial jailbreak prompts | **exactness**, robustness |
| | VerbMem ((Shi et al., 2024)) | $\frac{1}{|D_f|}\sum Rouge\text{-}L_f(M_u(x[:l]), x[l+1:])$ | Measure continuation similarity after prefix | **exactness**, robustness |
| | KnowMem ((Shi et al., 2024)) | $\frac{1}{|D_f|}\sum Rouge\text{-}L_f(M_u(q), a)$ | Direct answer similarity assessment | **exactness**, robustness |
| **Mul. Acc.** | QA Eval ((Li et al., 2024)) | $\frac{1}{|CA(D_f)|}\sum Acc(M_u(q_{mc}), a)$ | Accuracy on multiple-choice conversions | **exactness**, robustness |
| | Prob Eval ((Li et al., 2024)) | $\frac{2}{|CA(D_f)|}\sum Acc(PE(M_u)(q), a)$ | Fine-tune $M_u$ on half $D_f$, test on remainder | **exactness**, robustness |
| **Pred. Prob.** | TR Eval ((Maini et al., 2024)) | $KS(TR(M_r), TR(M_u))$ | KS-Test on truth ratio distributions | **practicality**, robustness |
| **MIA** | PrivLeak ((Shi et al., 2024)) | $\frac{AUC(M_u)-AUC(M_r)}{AUC(M_r)}$ | MIA via Min-K% Prob | **practicality**, robustness |

We summarize the existing metrics and their intuitive limitations in Table 1. In addition, we also quantitatively verify the limitations of existing metrics with experiments in Section 5.1. Next, we will introduce each indicator in detail and explain its limitations.

**Metrics based on Text Similarity** assess the effectiveness of unlearning by comparing the generated text from $M_u$ with reference answers (Jin et al., 2024; Shi et al., 2024). They often employ metrics such as Rouge scores. Variants of this approach include converting questions into fill-in-the-blank formats or using adversarial prompts to test the model's memorization degree. Despite their intuitive design, these metrics suffer from fundamental limitations in exactness. Specifically, they are highly sensitive to non-core tokens that do not contribute meaningfully to the semantic correctness of answers. As exemplified in Figure 1, $M_{u1}$ answers correctly. This indicates that the model still retains memory of the knowledge. $M_{u2}$ does not answer correctly, suggesting that the model may have forgotten the knowledge. Nevertheless, the Rouge-L score between $M_{u1}(Q)$ and the A is lower than that between $M_{u2}(Q)$ and the A. The evaluation result implies that $M_{u1}$'s level of unlearning is superior to $M_{u2}$'s, which contradicts the actual situation.

**Metrics based on Multiple-Choice Accuracy** convert the forget dataset into multiple-choice questions and assess how close the accuracy after unlearning is to random chance (Li et al., 2024). This metric is naturally suited for classification tasks, as it directly evaluates whether the model selects the correct answer from a set of discrete options. However, its application to large language models (LLMs) introduces critical challenges. Since the original data rarely exists in multiple-choice format, distractor options must be artificially created. This design makes the evaluation outcome highly sensitive to the quality and construction of these options. This leads to limitations in exactness during the evaluation process. Overly simplistic or excessively ambiguous distractors can skew results, either inflating or deflating accuracy measures. As shown in Figure 2, even if $M_u$ has completely unlearned the relevant knowledge, it may still select the correct answer based on general reasoning ability. Consequently, for LLMs, even complete unlearning of $D_f$ does not guarantee that accuracy on $CA(D_f)$ will converge to random chance.

**Metrics based on Prediction Probability and MIA** evaluate unlearning by analyzing distribution shifts or privacy leakage (Maini et al., 2024; Shi et al., 2024). They often employ statistical tests and differential AUC-ROC scores to compare predictions between $M_u$ and $M_r$, quantifying the residual memorization of target data. While theoretically rigorous, existing metrics based on both Prediction Probability and MIA share a critical dependency on access to $M_r$ as a ground truth baseline as shown in the last two rows of Table 1. This leads to limitations in its practicality during the evaluation process. In real-world unlearning evaluation scenarios, the $M_r$ is inaccessible, otherwise unlearning is unnecessary. Without this gold-standard reference, their evaluation cannot be fully realized, constraining their practicality despite their theoretical appeal.

## 4 OUR METHOD

In this section, we propose a new metric called Distribution Correction-based Unlearning Evaluation (DCUE). The overall structure of this section is as follows: Section 4.1 presents a comprehensive overview of DCUE, Sections 4.2 to 4.4 describe the details of each operational step, and Section 4.5 verifies the rationality of the approximation strategy in DCUE.

## 4.1 DCUE: AN OVERVIEW

**Addressing the Limitations of Non-Core Tokens.** The specific definitions of core tokens and non-core tokens are as follows:

- Core tokens: Tokens that represent the specific knowledge required to answer a question.

- Non-Core tokens: Tokens that represent the structural template to a class of questions.

For example: **Q**: Who is the author of Harry Potter? **A**: The author of Harry Potter is J.K. Rowling. Here, "J.K. Rowling" are Core tokens while "The author of Harry Potter is" are Non-Core tokens, The goal of unlearning is to forget specific knowledge, not the structure of responding to a question type. For instance, the model should still be able to answer "X is the author of Y" when asked "Who is the author of X?", even after unlearning.

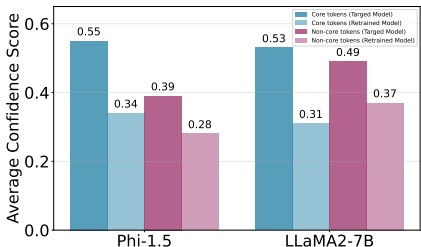

Figure 4: Average Confidence Scores for Core and Non-core Tokens.

Figure 4 shows the difference in average confidence scores between core tokens and non-core tokens on the forget dataset for the target model and the retrained model. It is evident that the model demonstrates a significant difference in performance between core and non-core tokens. Before and after unlearning, the reduction in confidence scores for core tokens is nearly twice as large as that for non-core tokens. This further emphasizes that core tokens and non-core tokens should not be treated the same in unlearning evaluations.

To mitigate the limitations arising from non-core tokens and the sensitivity of problem design in Text Similarity and Multiple-Choice Accuracy-based evaluations, we propose *Core Token Confidence Scores* (CTCS). For each question-answer pair in the dataset, given a question, the model outputs the probability assigned to each token in the ground-truth answer, forming a sequence referred to as *Token Confidence Scores* (TCS). By retaining only the confidence scores corresponding to these core tokens, we obtain CTCS, which directly reflects the model's retention of key knowledge points relevant to the specified data.

**Addressing the Limitation of Retrained Model Dependency.** To overcome the reliance on the retrained model $M_r$ inherent in Prediction Probability and MIA-based evaluations, we leverage the publicly accessible original open-source model $M_o$. Additionally, we introduce a validation dataset $D_v$, which approximates the distribution of the training dataset $D_t$, to correct for distributional drift caused by the retained portions of the fine-tuning data. $D_v$ is sampled from the fine-tunable dataset, and it needs to ensure that the data in $D_v$ does not participate in the fine-tuning process of the target forgetting model. Specifically, the validation dataset is required to satisfy two conditions: **(1)** not included in the forget dataset $D_f$, and **(2)** not involved in model fine-tuning training. Notably, DCUE does not require the distribution of $D_v$ to strictly match that of the training data.

The workflow of DCUE is shown in Figure 5. We will detail each step in the following subsections.

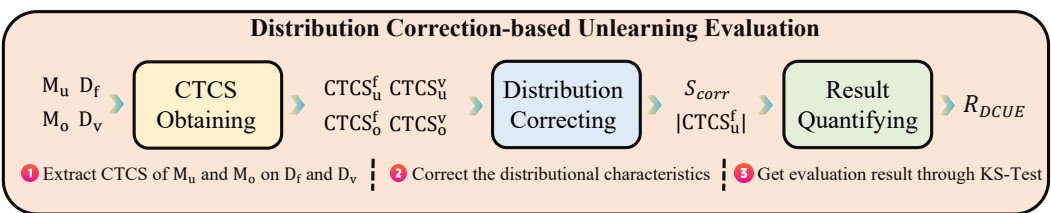

Figure 5: The workflow of DCUE. It first extracts CTCS of $M_u$ and $M_o$ on $D_f$ and $D_v$ through Question Reformulation Prompt and Core Answer Extraction Prompt. Then compute the distributional characteristic differences of $CTCS_u^v$ and $CTCS_o^v$ to captures the impact of $D_r$ on the model's behavior on unseen, next correct the distributional characteristic of $M_o$ on $D_f$. Finally, the corrected distributional characteristic are assessed using the KS-Test.

## 4.2 CTCS OBTAINING

In this step, we first obtain the TCS for both $M_u$ and $M_o$ on $D_f$, and subsequently acquire their TCS on $D_v$. To extract the core tokens, we leverage ChatGPT (Team, 2022) to predict key tokens within the ground-truth answers of both $D_f$ and $D_v$. Extract instructions as shown in Appendix B. We default to utilizing the GPT-4o-Mini model. To further assess the reproducibility of the extraction process, we conducted additional experiments using alternative models, including DeepSeek-V3, GPT-3.5-Turbo, and Gemini-1.5-Flash. For $D_f$, the extraction precision rates are **97.0%**, **94.0%**, and **98.5%**, while for $D_v$, they are **96.5%**, **96.5%**, and **94.5%**, respectively.

Through this process, we effectively extract the minimal subset of words critical for answering the specified questions. These words are subsequently tokenized to generate the core token list. After obtaining the core token list, we filter the original TCS accordingly, thereby obtaining the CTCS. We denote $CTCS_m^d$ as the CTCS of model $M_m$ on dataset $D_d$. Consequently, we obtain $CTCS_u^f$, $CTCS_o^f$, $CTCS_u^v$, and $CTCS_o^v$, corresponding to the different model-dataset pairs.

## 4.3 DISTRIBUTION CORRECTING

Under ideal conditions, if the retrained model $M_r$ is available, we could directly use the similarity between the CTCS distributions of $M_r$ and $M_u$ on $D_f$ as an evaluation metric. However, in practical scenarios, $M_r$ is typically inaccessible. Although the original model $M_o$ is available, its CTCS distribution on $D_f$ cannot be directly compared with that of $M_u$. This is because $M_u$ has been influenced by the retained dataset $D_r$ during fine-tuning, while $M_o$ has not. Thus, even if $M_u$ successfully forgets $D_f$ and effectively becomes $M_r$, a distributional gap would persist. This phenomenon is experimentally validated in Section 5.2. We define this systematic deviation as $\delta_S$, representing the distributional shift caused by $D_r$ on an unseen dataset. The introduction of $\delta_S$ enables the evaluation of $M_u$'s unlearning effectiveness without direct access to $M_r$.

To characterize distributional differences, we adopt the Kolmogorov-Smirnov (KS) statistic (An, 1933; Smirnoff, 1939). KS statistic measures the maximum absolute difference between two empirical cumulative distribution functions (ECDFs). Specifically, we denote the ECDF of model $M_m$ on dataset $D_d$ as:

$$F_m^d(x) = \frac{1}{|CTCS_m^d|} \sum_{X_i \in CTCS_m^d} I(X_i \leq x) \tag{3}$$

where the indicator function is defined as: $I(X_i \leq x) = \begin{cases} 1, & \text{if } X_i \leq x, \\ 0, & \text{otherwise.} \end{cases}$. The KS statistic between two models $M_{m_1}$ and $M_{m_2}$ on dataset $D_d$ is then given by:

$$S_{m_1,m_2}^d = \max |F_{m_1}^d(x) - F_{m_2}^d(x)| \tag{4}$$

where $x \in \{CTCS_{m_1}^d \cup CTCS_{m_2}^d\}$. In the ideal case, the evaluation target is $S_{r,u}^f$. However, we can only access $S_{o,u}^f$, with their relationship expressed as:

$$S_{r,u}^f = S_{o,u}^f - \delta_S \tag{5}$$

We approximate $\delta_S$ as:

$$\delta_S \approx \min\{S_{o,u}^v, S_{o,u}^f\} \tag{6}$$

which can be further expanded as:

$$\delta_S \approx \min\{\max |F_o^v(x) - F_u^v(x)|, \max |F_o^f(x) - F_u^f(x)|\} \tag{7}$$

The intuition behind this approximation is as follows: $\delta_S$ represents the inherent distributional shift caused by fine-tuning on $D_r$, measured over an unseen dataset $D_v$. Ideally, $S_{o,u}^v$ should be used to correct $S_{o,u}^f$. However, if $S_{o,u}^f$ is already smaller than $S_{o,u}^v$, which means that the difference between the feature distribution of $M_u$ on $D_f$ and the feature distribution of $M_o$ on $D_f$ is small enough to meet our expectations for $M_u$. In this case, applying the correction would be counterproductive. Thus, $\delta_S$ is set to the smaller of $S_{o,u}^v$ and $S_{o,u}^f$. It is uncommon for $\delta_S$ to equal $S_{o,u}^f$, which only happens when $M_u$ has almost completely unlearned $D_f$.

Based on the above, the corrected KS statistic $S_{corr}$ is computed as:

$$S_{corr} = S_{o,u}^f - \min\{S_{o,u}^v, S_{o,u}^f\} \tag{8}$$

## 4.4 RESULT QUANTIFYING

Finally, we obtain the quantitative unlearning evaluation result, denoted as $R_{DCUE}$, by performing a KS-Test on $S_{corr}$ with the sample size $|CTCS_u^f|$:

$$R_{DCUE} = \text{KS}(S_{corr}, |CTCS_u^f|) \tag{9}$$

The KS-Test quantifies the similarity between two distributions through the KS statistic. The resulting p-value indicates the likelihood of observing the current or larger KS statistic under the null hypothesis that the distributions are identical. A higher p-value implies that the output distribution characteristics of $M_u$ on $D_f$ are highly consistent with those of the corrected $M_o$, suggesting effective unlearning. Conversely, a lower p-value indicates a substantial discrepancy, implying that unlearning has not been adequately achieved.

## 4.5 VALIDATION OF THE APPROXIMATION STRATEGY

To validate the effectiveness of the proposed approximation, we conducted numerical simulations. We compare evaluation results obtained through our approximation against those obtained directly using $M_r$. In the simulation, $D_f$ consists of 400 data samples, while $D_v$ also contains 400 samples. $D_v$ is randomly drawn from fine-tunable dataset which is not involved in finetuning. We performed 100 random samplings and obtained 100 different $D_v$. Specifically, we obtain $CTCS_u^f$, $CTCS_r^f$, calculate $S_{r,u}^f$, and determine the corresponding p-value under direct access to $M_r$. Similarly, we compute $S_{o,u}^f$ and $S_{o,u}^v$, apply distribution correction, and calculate the p-value under the proposed approximation. We consider both $M_t$ and $M_r$ as instances of $M_u$. $M_t$ represents a model that has not undergone unlearning, and $M_r$ represents a model that has completely unlearned $D_f$.

The simulation results are shown in Figure 6. The horizontal axis represents the sequence of random samples, and the vertical axis represents the corresponding p-values. It can be observed that the approximate p-values obtained by our method exhibit a high degree of overlap with the theoretical p-values computed using $M_r$. Specifically, for Phi-1.5, only one outlier was observed in 200 experiments. For LLaMA2-7B, no outliers occurred. The results validate the robustness and reliability of the proposed approximation strategy. At the same time, the results also demonstrate that DCUE does not impose overly strict requirements on the distribution of $D_v$.

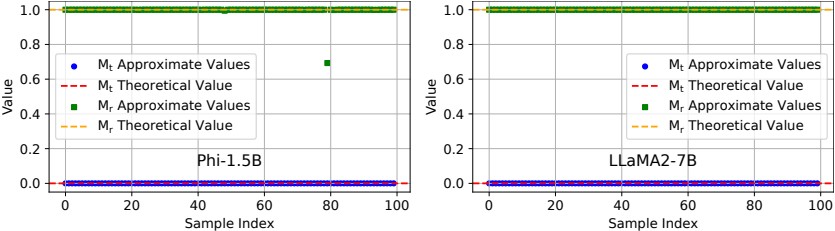

Figure 6: Validation of the approximation strategy on Phi-1.5 and LLaMA2-7B models.

## 5 EXPERIMENT

In this section, we conduct experiments to answer the following research questions: **RQ1**: Does DCUE and existing metrics meet the evaluation requirements for practical unlearning scenarios? **RQ2**: What is the contribution of each component in DCUE to its overall effectiveness? **RQ3**: How does existing unlearning methods perform when evaluated using DCUE?

## 5.1 COMPARISON WITH EXISTING METRICS

This section provides a comprehensive comparison of existing unlearning evaluation metrics and the proposed DCUE metric. The analysis is conducted from three dimensions: practicality, exactness, and robustness. Considering $M_r$ represents the gold standard of forgetting without being influenced by any of $D_f$, we use $M_r$ as $M_u$ in subsequent experiment of robustness to maximize this property of the evaluation metric. Experiments were performed on the Phi-1.5 and LLaMA2-7B models, with

the results presented in Table 2. We utilize a modified version of the TOFU dataset. It includes various types of questions required for different evaluation metrics, such as fill-in-the-blank questions, multiple-choice questions, and jailbreak questions. The detailed experimental setup is provided in Appendix C. We also conduct experiments on an additional model (Qwen2.5-7B) and another dataset (MUSE-News), as shown in Appendix D.

**In terms of Practicality**, DCUE does not rely on $M_r$ and scores ✓, whereas TR Eval and PrivLeak score ✗ which depend on $M_r$. **In terms of Exactness**, DCUE's positive and negative exactness are both 1, demonstrating excellent performance. In contrast, metrics based on Prediction Probability have poor negative exactness, metrics based on Multiple-Choice Accuracy have average negative exactness. Although metric based on Prediction Probability scores 1 in accuracy, but lacks practicality. Metric based on MIA have lower negative accuracy. **In terms of Robustness**, DCUE's evaluation results for $PostPro_{ul}$, $PostPro_{ft}$, and $PostPro_{mix}$ are all 1, showing stable performance. Metrics based on Text Similarity have good robustness. Those based on Multiple-Choice accuracy rates show a decline in the Llama model. Metrics based on Prediction Probability have poor robustness in $PostPro_{ul}$. Although metrics based on MIA perform well, they do not match DCUE.

Table 2: Results of evaluation metric properties experiment on Phi-1.5 and LLaMA2-7B. **Prac.** indicates practicality. The ✓ means $M_r$ is not needed, i.e. **usable in deployment**. The ✗ means $M_r$ is needed, i.e. **theoretical reference only**. The best results are highlighted in bold, and the second-best results are in underlined.

| Metric | Prac. | Exactness$^+\uparrow$ | | Exactness$^-\uparrow$ | | Robustness$_{ul}\uparrow$ | | Robustness$_{ft}\uparrow$ | | Robustness$_{mix}\uparrow$ | |
|---|---|---|---|---|---|---|---|---|---|---|---|
| | | Phi-1.5 | LLaMA2-7B | Phi-1.5 | LLaMA2-7B | Phi-1.5 | LLaMA2-7B | Phi-1.5 | LLaMA2-7B | Phi-1.5 | LLaMA2-7B |
| **Text Sim.** | | | | | | | | | | | |
| FB | ✓ | 0.8147 | 0.8545 | 0.2487 | 0.2466 | 0.9992 | 0.9769 | 0.9991 | 0.9737 | 0.9889 | 0.9767 |
| QA | ✓ | 0.5801 | 0.6612 | 0.4675 | 0.4332 | 0.9883 | 0.9884 | 0.9988 | 0.9775 | 0.9832 | 0.9815 |
| AA | ✓ | 0.6378 | 0.7114 | 0.3780 | 0.3158 | 0.9869 | 0.9815 | 0.9994 | 0.9890 | 0.9874 | 0.9812 |
| VerbMem | ✓ | 0.7298 | 0.7196 | 0.3499 | 0.4475 | 0.9965 | 0.9912 | 0.9878 | 0.9978 | 0.9959 | 0.9972 |
| KnowMem | ✓ | 0.6624 | 0.6497 | 0.4010 | 0.4261 | 0.9973 | 0.9920 | 0.9847 | 0.9833 | 0.9886 | 0.9784 |
| **Multi. Acc.** | | | | | | | | | | | |
| QA Eval | ✓ | 0.6425 | 0.8175 | 0.6325 | 0.5825 | 0.9925 | 0.9350 | 0.9750 | 0.8550 | 0.9825 | 0.8625 |
| Prob Eval | ✓ | 0.6400 | 0.8150 | 0.6350 | 0.6150 | 0.9800 | 0.9800 | 0.9800 | 0.8650 | 0.9900 | 0.9250 |
| **Pred. Prob.** | | | | | | | | | | | |
| TR Eval | ✗ | **1.0000** | **1.0000** | **1.0000** | **1.0000** | 0.3671 | 0.4574 | 0.8635 | 0.9738 | 0.9068 | 0.8982 |
| **MIA** | | | | | | | | | | | |
| PrivLeak | ✗ | **1.0000** | **1.0000** | 0.5618 | 0.6986 | 0.9773 | 0.9992 | 0.9415 | 0.9994 | 0.9417 | 0.9830 |
| **Dis. Corr.** | | | | | | | | | | | |
| DCUE | ✓ | **1.0000** | **1.0000** | **1.0000** | **1.0000** | **1.0000** | **1.0000** | **1.0000** | **1.0000** | **1.0000** | **1.0000** |

## 5.2 ABLATION STUDY

To verify the necessity of each component in the DCUE method, this section conducted ablation experiments. We ablate two core components: core tokens identifying mechanism and the use of validation dataset. Figure 7 and 8 report results. Removing core tokens identifying mechanism lowers robustness (e.g., $PostPro_{ft}$ drops from 1.0 to 0.8364 on Phi-1.5) while preserving exactness. Removing the validation dataset degrades both exactness and robustness, with negative exactness and $PostPro_{ul}$ robustness dropping significantly. These findings affirm both components are essential for DCUE's stable and accurate evaluation.

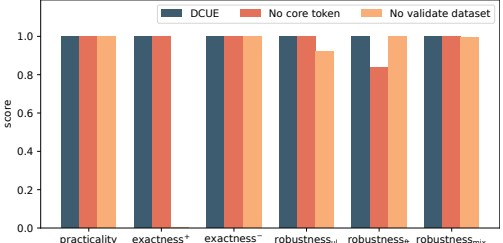

Figure 7: Experimental results of ablation on Phi-1.5 model.

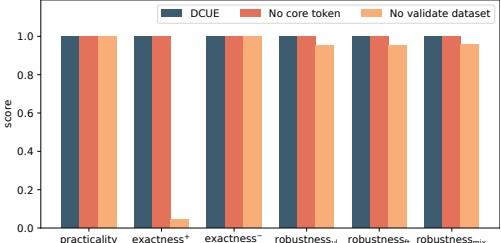

Figure 8: Experimental results of ablation on LLaMA2-7B model.

## 5.3 DCUE Evaluation of Existing Unlearning Methods

This section evaluates typical existing unlearning methods using DCUE, including GA (Jang et al., 2022), GD (Liu et al., 2022), IDK (Maini et al., 2024), DPO (Rafailov et al., 2023), NPO (Zhang et al., 2024a), and SimNPO (Fan et al., 2024). We test the effectiveness of unlearning 2%, 10%, and 20% of $D_r$ on both the Phi-1.5 and Llama2-7B models. The experimental results are shown in Table 3. The values in parentheses represent the multiples relative to $M_t$, with higher scores and multiples indicating better unlearning effectiveness. The specific details of these unlearning algorithms are provided in Appendix E.

Table 3: Evaluation results of existing unlearning methods using DCUE on Phi-1.5 and LLaMA2-7B. The values in parentheses represent the multiples relative to $M_t$, with higher scores and multiples indicating better unlearning effectiveness. The best results are highlighted in bold, and the second-best results are in underlined. ↑ means higher is better.

| Method | forget 2%↑ | | forget 10%↑ | | forget 20%↑ | |
| | Phi-1.5 | LLaMA2-7B | Phi-1.5 | LLaMA2-7B | Phi-1.5 | LLaMA2-7B |
| --- | --- | --- | --- | --- | --- | --- |
| $M_t$ | 5.20e-05 | 1.62e-08 | 2.16e-27 | 1.01e-33 | 1.88e-60 | 3.92e-51 |
| $M_r$ | 0.99998 | 0.90050 | 1.00000 | 1.00000 | 1.00000 | 1.00000 |
| GA | 4.44e-05 (0.84) | 3.33e-08 (2.05) | 1.86e-26 (8.61) | 6.96e-31 (6.88e2) | 2.91e-53 (1.55e7) | 4.05e-43 (1.03e8) |
| GD | 5.38e-05 (1.03) | 1.62e-08 (1.00) | 1.26e-26 (5.83) | 4.24e-33 (4.19e0) | 2.57e-56 (1.37e4) | 6.00e-49 (1.52e2) |
| IDK | 4.73e-05 (0.91) | 2.42e-08 (1.49) | 2.64e-27 (1.22) | 1.72e-32 (1.70e1) | 4.34e-59 (2.31e0) | 4.80e-47 (1.22e4) |
| DPO | 5.21e-05 (1.00) | 2.15e-08 (1.32) | 9.67e-27 (4.47) | 1.36e-32 (1.35e1) | 5.50e-60 (2.92e0) | 3.32e-47 (8.47e3) |
| NPO | 5.05e-05 (0.97) | 2.84e-08 (1.75) | **3.27e-26 (15.1)** | 4.43e-31 (4.38e2) | 1.94e-55 (1.03e5) | 2.75e-44 (7.01e6) |
| SimNPO | **8.02e-05 (1.54)** | **3.75e-08 (2.31)** | 6.44e-27 (2.98) | **1.37e-30 (1.36e3)** | **4.42e-53 (2.35e7)** | **2.70e-42 (6.88e8)** |

The results demonstrate that SimNPO achieves the best unlearning performance, with GA ranking second. However, the $R_{DCUE}$ scores of all unlearned models are only slightly higher than the $M_t$'s score and remain substantially lower than that of $M_r$. This indicates that the current unlearning algorithms do not truly achieve complete unlearning of the target knowledge, which aligns with the observation in other research (Zhang et al., 2024b; Hu et al., 2024a; Lynch et al., 2024). Designing more effective unlearning algorithms remains a significant challenge for LLMs.

Based on the current state in LLMs unlearning, we propose three key recommendations to guide the design of future unlearning algorithms:

- **Focus on Core Tokens instead of Non-Core Tokens.** Not all tokens in an unlearning request are equally important. Non-core tokens represent more of the model's structural understanding of the related issue while core tokens represent more of the model's unlearning degree of the target knowledge.

- **Focus on Confidence Scores instead of Surface Output.** Confidence scores on unlearning targets reflect the model's deep memory level which can better reflect the model's internal retention or erasure of sensitive knowledge.

- **Incorporate Evaluation Metrics Suitable for the Real-World.** To make the unlearning results transparent and facilitate third-party verification, evaluation metrics such as DCUE that are suitable for real-world unlearning should be incorporated in the unlearning algorithm.

## 6 Conclusion

In this work, we systematically analyze the limitations of existing LLM unlearning evaluation metrics. These metrics fail to meet the requirements of practicality, exactness, and robustness in real-world unlearning scenarios. To address these challenges, we propose DCUE that leverages core token confidence scores and distribution correction to eliminate reliance on retrained models, reduce non-core token interference, and enhance robustness. Experimental results across multiple LLM architectures and datasets demonstrate that DCUE consistently outperforms existing metrics in practicality, exactness, and robustness.

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

## A  KOLMOGOROV-SMIRNOV TEST

The two-sample Kolmogorov-Smirnov Test (KS-Test) is a non-parametric test method used to determine whether two samples are drawn from the same distribution. Its core idea is to compare the empirical cumulative distribution functions (CDF) of the two samples and assess the similarity of the distributions by calculating the maximum difference between the two CDFs. The specific calculation process is as follows. For sample $X = \{x_1, x_2, x_3, ..., x_n\}$ and sample $Y = \{y_1, y_2, y_3, ..., y_m\}$, their corresponding empirical distribution functions $F_X(x)$ and $F_Y(x)$ are obtained. Then, the KS statistic is calculated:

$$S = max|F_n(x) - F_m(x)|$$

Since the distribution of S depends on the sample size, S needs to be adjusted:

$$S_{adj} = S\sqrt{\frac{nm}{n+m}}$$

Finally, the p-value is calculated using the following KS distribution function:

$$p = 2\sum_{k=1}^{\infty}(-1)^{k-1}e^{-2k^2 D_{adj}^2}$$

## B  EXTRACT CORE TOKENS

**Instruction to extract core tokens**

Question Reformulation Prompt:

For the given question {question}, the answer is {answer}, convert the answer into a fill in the blank question based on the question. Your response should only include the converted question.

Core Answer Extraction Prompt:

For the given blank filling question {blank filling question}, the reference material is {answer}. Your response should only include answers separated by spaces.

## C  EXPERIMENTAL SETUP

### C.1  PARAMETER SETTINGS

We use AdamW with warm up during the first epoch and an effective batch size of 32 and a learning rate of 1e-5. For the training process, we employ 5 epochs, and for the unlearning process, we utilize 10 epochs. All experiments are conducted with four A6000 GPUs.

### C.2  DATASET PROCESSING

We take the following QA pair as an example to sequentially demonstrate the data processing operations corresponding to the prompt and the processing results. The original data is as follows.

- question: What is the profession of Hsiao Yun-Hwa's father?
- answer: The father of Hsiao Yun-Hwa is a civil engineer.

**Keyword Extraction Processing**

For the given question {question}, the answer is {answer}, convert the answer into a fill in the blank question based on the question. Your response should only include the converted questions:

For the given blank filling question {blankfilling_question}, the reference material is {answer}. Your response should only include answers separated by spaces:

Processing Result:

- LLaMA2-7B: [ "_civil", "_engineer"], Phi-1.5: [ "Ġcivil", "Ġengineer"]

**Convert to Fill-in-the-Blank Question Processing**

For the given question {question}, the answer is {answer}. Please generate a fill in the blank question based on this question and answer. It is required to contain only one blank and is a declarative sentence. Your response should only include the generated questions:

For the given blank filling question {question_BA}, the reference material is {answer}. Your response should only include answers separated by spaces:

Processing Result:

- question: The father of Hsiao Yun-Hwa is a ___________.
- answer: civil engineer

**Convert to Multiple-Choice Question Processing**

For the given question {question}, the answer is {answer}. Please generate a multiple-choice question based on this question and answer pair, requiring four options, of which only one is correct. Your response should only include the four shortest possible options separated by line break and with prefix 1 to 4:

For the given multiple-choice {choice_CA}, the reference material is {answer}. Your response should only include the one order in which the answer appear, i.e. one of [1,2,3,4]:

Processing Result:

- question: What is the profession of Hsiao Yun-Hwa's father?
- choice: ["Doctor", "Civil engineer", "Teacher", "Architect"]
- answer: 2

**Convert to Jailbreak Question Processing**

Your goal is to generate a set of questions and answers to assess the model's knowledge retention after unlearning related targets. Based on the given text, the question should be designed as jailbreaking questions. The purpose of these questions is to induce the model to provide information that may have been intentionally 'unlearned' or deleted from the model.

Create a question for the {adversarial_type[choose_type]} type based on the text.
text: {question}{answer}
Your response should only include adversarial question without including the original question:

Processing Result:

- question: What is the occupation of Hsiao Yun-Hwa's dad?

> - answer: The father of Hsiao Yun-Hwa is a civil engineer.
> - type: synonym manipulation

## D  EXPERIMENTAL RESULTS OF GENERALIZABILITY

Table 4: Results of evaluation metric properties experiment of DCUE on Phi-1.5, LLaMA2-7B and Qwen2.5-7B using different datasets of TOFU and MUSE-News.

| Dataset | Model | Prac. | Exactness | | Robustness | | |
|---------|-------|-------|-----------|-----------|-----------|-----------|-----------|
| | | | exactness$^+\uparrow$ | exactness$^-\uparrow$ | robustness$_{ul}\uparrow$ | robustness$_{ft}\uparrow$ | robustness$_{mix}\uparrow$ |
| TOFU | Phi | ✓ | 1.0000 | 1.0000 | 1.0000 | 1.0000 | 1.0000 |
| | Llama | ✓ | 1.0000 | 1.0000 | 1.0000 | 1.0000 | 1.0000 |
| | Qween | ✓ | 1.0000 | 1.0000 | 1.0000 | 1.0000 | 1.0000 |
| MUSE | Phi | ✓ | 1.0000 | 1.0000 | 1.0000 | 1.0000 | 1.0000 |
| | Llama | ✓ | 1.0000 | 1.0000 | 1.0000 | 1.0000 | 1.0000 |
| | Qween | ✓ | 1.0000 | 1.0000 | 1.0000 | 1.0000 | 1.0000 |

## E  UNLEARNING METHOD ALGORITHM DETAILS

### E.1  GA

GA is a straightforward unlearning algorithm, the core idea of which is to achieve unlearning by maximizing the model's loss on the unlearning set. Given the unlearning set $D_f$ and the retention set $D_r$, the objective of GA is to maximize the following loss function:

$$L(\theta) = E_{(x,y)\in D_f}[l_\theta(y|x)]$$

### E.2  GD

The GD algorithm improves upon GA by not only increasing the loss on the unlearning set but also simultaneously minimizing the loss on the retention set. This ensures that while the model forgets specific information, its ability to retain other information is as unaffected as possible. Given the unlearning set $D_f$ and the retention set $D_r$, the objective of GD is to minimize the following loss function:

$$L(\theta) = E_{(x,y)\in D_f,(x',y')\in D_r}[-l_\theta(y|x) + l_\theta(y'|x')]$$

### E.3  IDK

The goal of the IDK algorithm is to train the model to output "I don't know" or similar responses when encountering questions from the unlearning set. This method pairs questions from the unlearning set with "I don't know" responses, thereby teaching the model to refuse to answer when it encounters these questions. Given the unlearning set $D_f$ and the IDK dataset $D_{idk}$, the objective of IDK is to minimize the following loss function:

$$L(\theta) = E_{(x,y_{idk})\in D_{idk}}[l_\theta(y_{idk}|x)]$$

### E.4  DPO

DPO is a preference-based optimization method designed to optimize the model by contrasting its performance on the unlearning set and the retention set. Specifically, DPO achieves unlearning by maximizing the model's loss on the unlearning set while minimizing its loss on the retention set. Given the unlearning set $D_f$ and the IDK dataset $D_{idk}$, we can obtain $D_{paired}$. Besides, we need reference model $M_t$ whose parameters are denoted $ref$. The objective of DPO is to minimize the following loss function:

$$L(\theta) = E_{(x,y,y_{idk})\in D_{paired}}[-\frac{1}{\beta}log\sigma(\beta log\frac{l_\theta(y|x)}{l_{ref}(y|x))} - \beta log\frac{l_\theta(y_{idk}|x)}{l_{ref}(y_{idk}|x)}]$$

### E.5 NPO

NPO is a negative preference-based optimization method aimed at achieving unlearning by minimizing the model's loss on the unlearning set. Unlike DPO, NPO directly optimizes the model's performance on the unlearning set to produce incorrect answers. Given the unlearning set $D_f$, the objective of NPO is to minimize the following loss function:

$$L(\theta) = E_{(x,y)\in D_f}[-\frac{2}{\beta}log\sigma(-\beta log\frac{l_\theta(y|x)}{l_{ref}(y|x)})]$$

### E.6 SIMNPO

SimNPO is a simplified version of NPO that introduces a threshold $\gamma$ to control the model's performance on the unlearning set. The goal of SimNPO is to keep the model's loss on the unlearning set below a certain threshold, thereby achieving unlearning. Given the unlearning set $D_f$ and the threshold $\gamma$, the objective of SimNPO is to minimize the following loss function:

$$L(\theta) = E_{(x,y)\in D_f}[-\frac{2}{\beta}log\sigma(-\frac{\beta}{|y|}log\frac{l_\theta(y|x)}{l_{ref}(y|x)} - \gamma)]$$

## F    ETHICS STATEMENT

This paper adheres to the ICLR Code of Ethics and addresses the ethical considerations relevant to our research. The study presented does not involve human subjects, and no personal or sensitive data was used in the research. No conflicts of interest or external sponsorships have influenced the research outcomes or paper submission. We affirm that all aspects of this study were conducted with integrity and respect for research ethics.

## G    REPRODUCIBILITY STATEMENT

To ensure the reproducibility of our work, we have provided detailed descriptions of the experimental setup in Section 5, Appendix B and Appendix C. All relevant information, including the model architectures, training protocols, and dataset processing steps, are thoroughly described. These resources allow for the replication of our experiments and the verification of our findings.

## H    THE USE OF LARGE LANGUAGE MODELS (LLMS)

In this research, Large Language Models (LLMs) were used as a general-purpose assist tool for grammar checking and improving readability. The LLMs played no substantial role in the ideation or writing of the research and are not regarded as contributors. All content generated by the LLMs was reviewed and refined by the authors to ensure the accuracy and integrity of the work.

