# OpenReview forum: "Towards Evaluation for Real-World LLM Unlearning"
_ICLR.cc/2026/Conference — ICLR 2026 Conference Withdrawn Submission_

### Official Review · Reviewer_5gcj · 2025-10-27

**Soundness:** 2
**Presentation:** 3
**Contribution:** 1
**Rating:** 2
**Confidence:** 3

**Summary:**

The paper “Towards Evaluation for Real-World LLM Unlearning” focuses on the critical but underexplored problem of evaluating large language model (LLM) unlearning in realistic deployment settings. The authors point out that existing evaluation metrics—such as those based on text similarity, multiple-choice accuracy, prediction probability, or membership inference attacks—often fail in terms of practicality, exactness, and robustness. To address these issues, the paper introduces a new evaluation framework called Distribution Correction-based Unlearning Evaluation (DCUE).

**Strengths:**

DCUE works without access to a retrained baseline, aligning with real-world deployment constraints.

**Weaknesses:**

There exist numerous prior studies addressing the same problem, many of which were proposed one or two years earlier. It is therefore difficult to identify clear novelty or significance in this paper. In comparison, the contribution appears incremental and considerably weaker than that of the referenced works.

[1] Position: LLM Unlearning Benchmarks are Weak Measures of Progress

[2] BLUR: A Benchmark for LLM Unlearning Robust to Forget-Retain Overlap

[3] OpenUnlearning: Accelerating LLM Unlearning via Unified Benchmarking of Methods and Metrics

[4] Unlearning with Control: Assessing Real-world Utility for Large Language Model Unlearning

The paper is limited to the unlearning of fine-tuning data, as a paper to propose metric, it is clear that the proposed methods are not general enough.

Practicality, exactness, and robustness have all been mentioned in previous works, while the authors have not referred some related paper. Hence, I do not think it is a mature and professional draft.

The paper suggests another set of metrics to evaluate existing metrics, how to ensure they (i.e. eqs. 1 -2)  are reliable? How to ensure that they are not crafted to bias to the author proposed metrics. Discussions like Sec 3.2 are also appear in many previous works, like [3]. I cannot find something new from the authors’ discussion. Also, some more recent metrics, as well as LaaJ, are not mentioned.

The costs of the proposed evaluation framework is high. It require LLMs to separate core and non-core tokens. The authors have not proved if LLMs are reliable in finding those key tokens. Even so, why don’t we directly use LaaJ, it seems more reasonable and direct.

MIA-based evaluations and Prob do not always reliable on retrained models, the authors somehow overclaim it. Also, does confidence score reliable in evaluating unlearning? It seems that it is very sensitive to attacks, even for those key tokens.

What’s the experimental setup of Table 2? Did the authors conduct experiments with GA or some other methods? Anyway, the authors want to show their superiority,  but under very limited number of methods, experimental setups, and backbones, making it less convincing to show the reliability of the proposed methods.

The experiments are limited to limited number of experiments, and I cannot draw something new from the results. The summary at the end of Sec 5.3 is also not very attractive.

**Questions:**

Please see the weakness above

---

### Official Review · Reviewer_fJYU · 2025-10-30

**Soundness:** 2
**Presentation:** 3
**Contribution:** 2
**Rating:** 4
**Confidence:** 5

**Summary:**

This paper focuses on the problem of real-world unlearning, identifying the limitations of existing evaluation metrics and exploring more accurate and practically relevant standards for assessing model unlearning performance. The core contribution lies in the introduction of a new metric, DCUE, designed to address several key deficiencies in current approaches.

    1.Independence from external reference models: Recognizing that a gold-standard model is often unavailable in real-world scenarios, the paper proposes a metric that does not rely on such an external reference.

    2.Token-level discrimination: Existing metrics fail to distinguish between key and non-key tokens, leading to biased evaluation results. DCUE explicitly considers this issue and introduces a simple yet effective prompt-based solution.

    3.Robustness to post-operation changes: Many current metrics are sensitive to post-unlearning perturbations. DCUE mitigates this problem by incorporating a KS-test–based design, which enhances robustness and stability.

Overall, these ideas provide a valuable and timely perspective on an important challenge faced by current LLM unlearning research.

However, despite addressing an important problem, the paper still exhibits clear limitations in terms of novelty, experimental design, and overall contribution. The specific strengths and weaknesses are discussed in detail in the following parts.

**Strengths:**

This work addresses an important and timely topic concerning the evaluation metrics for LLM unlearning. It is true that conventional NLP metrics may not adequately reflect a model’s actual performance after unlearning, and highlighting this gap is a meaningful contribution.

DCUE demonstrates effectiveness by explicitly accounting for key tokens and robustness to post-unlearning variations. The approach is well aligned with human judgment, making it a more practical and realistic evaluation metric compared to prior methods.

The experimental results across multiple settings provide partial evidence supporting the reliability and effectiveness of DCUE.

The paper is well written and clearly organized, making it easy to follow the main ideas and experimental procedures.

**Weaknesses:**

1. From a motivational standpoint, DCUE is inspired by the recognition of key tokens as critical elements in unlearning evaluation. Nevertheless, several relevant studies [1][2][3] that have discussed the importance of key tokens are missing from the references. Although this does not diminish DCUE’s originality as a metric incorporating key-token effects, acknowledging those works would provide a more complete and accurate positioning of the paper. These prior studies explored the role of key tokens mainly in the context of optimization rather than evaluation, and thus are conceptually related but distinct.

2. Given that this paper primarily proposes a new evaluation metric, the lack of comparison with several advanced existing metrics [4,5] is difficult to justify. This omission is particularly problematic since those metrics have been demonstrated in subsequent work to represent the current state of the art.

3. Regarding the experimental setup—particularly the experiments based on the modified TOFU dataset—there are potential concerns about fairness. Modifying a benchmark can introduce risks of result-oriented optimization through selective data choices. Therefore, I strongly suggest providing a clearer description of the experimental setup, explicitly detailing whether the data selection process was randomized. It would also be helpful to report the mean and variance of results across multiple random samples to ensure the validity and reproducibility of the findings.

4. The overall contribution of the paper is rather narrow, as it focuses solely on introducing a new metric. As a result, the work feels limited in scope and may not meet the level of contribution expected for publication at ICLR. Based on prior studies, incorporating even a simple optimization approach that improve unlearning performance on the DCUE metric would make the paper considerably stronger and more complete.


[1] Not All Tokens Are Meant to Be Forgotten. Arxiv 2506.03142

[2] Not Every Token Needs Forgetting: Selective Unlearning to Limit Change in Utility in Large Language Model Unlearning.  Arxiv 2506.00876

[3] Exploring Criteria of Loss Reweighting to Enhance LLM Unlearning. ICML2025

[4] Towards Effective Evaluations and Comparisons for LLM Unlearning Methods. ICLR2025

[5] OpenUnlearning: Accelerating LLM Unlearning via Unified Benchmarking of Methods and Metrics. NIPS2025 DB track

**Questions:**

Please refer to the Weakness.

I would be willing to increase my score if the authors can adequately address the concerns and clarify the issues discussed above.

---

### Official Review · Reviewer_DP8d · 2025-10-30

**Soundness:** 1
**Presentation:** 1
**Contribution:** 2
**Rating:** 2
**Confidence:** 4

**Summary:**

This paper conceptualizes successful machine unlearning in LLMs as requiring three properties:
- Practicality: the ability of a measure to be evaluated without expensive retraining
- Exactness: how well a measure can correctly reflect the degree of unlearning
- Robustness: reliability under modifications to the model
They propose a distributional method (DCUE) to evaluate unlearning by identifying core tokens, estimating baselines for how likely they should be, and evaluating distributional differences using KS distance.

**Strengths:**

S1: I think that this paper gets the most important thing right: observing the issues with current evals and working to fix them. Although I think that table 1 needs much more explanation thatn it current has (See below), I believe that it reflects a major truth about unlearning evals all having problems.

S2: I think that the end of section 5 gives needed and unarguably correct wisdom. Framing the paper around that core idea was a good choice.

**Weaknesses:**

W1: I don't think that practicality, exactness, and robustness capture all the desiderata from unlearning well. Unlearning means different things in different contexts. Some people define it as erasing the influence of data. Others as removing a capability. In different cases, people care about robustness to different types of adversarial and non-adversarial manipulations to inputs or model parameters. I don't think that this paper is making useful conceptual progress by broadly defining the objectives of unlearning in the way that it does. I am also not sure if exactness and robustness are entirely distinct. All three seem somewhat underspecified.

W2: The sentence on line 72-74 seems to be something that almost all unlearning evaluations do. So this does not seem to me to be a contribution of the paper. As I read the introduction, I am also not sure what "core token" means. I work in unlearning, but I don't recall ever coming across this term. Meanwhile, I think that the KS test does a relatively poor job of encapsulating what people generally refer to when they care about "robustness".

W3: I am fairly skeptical of the author's assumption that the distribution matching method they use is sufficient to achieve exactness. One could easily come up with cases in which it would not, such as when I unlearn some text that has informational overlap with other things in the model's training data. Overall, it seems like the argument in the paper that DCUE satisfies the three key desiderata while other methods do not is based on ad hoc approximation in section 4.5 instead of comparison to gold-standard retraining.

W4: I didn't spot a justification for using KS over another method.

W5: Section 4.5 introduces a method for evaluating evaluation methods under the assumption that the evaluation in 4.5 reflects some kind of gold standard. But if so, why not just evaluate unlearning methods direclty using the approximation from 4.5?

Writing things:
- The abstract uses about 6 terms that aren't clear because they aren't explained in context. The abstract is not currently able to serve as an effective standalone summary of the paper.
- This paper isn't clear on how unlearning is defined. Different parts of the literature define it in terms of a counterfactual based on the training data or a behavioral change in what the network learns...I now see this is defined in section 2, but I think the writing of the abstract/intro could be updated to be clear about what the goal is.
- The writing of the paper seems relatively unclear, maybe reflecting limited experience. For example, terms are not always defined when they are introduced. Figures contain unexplained ancronyms and captions insufficient to convey the key ideas on their own. The abstract left me with more questions than answers. Some claims like why some methods have the limitations they do in table 1 aren't explained. I am inclined to say that from an overall writing clarity standpoint alone, this paper needs more work before it's ready to be out.

**Questions:**

See weaknesses above

---

### Official Review · Reviewer_JDM9 · 2025-11-01

**Soundness:** 3
**Presentation:** 2
**Contribution:** 2
**Rating:** 4
**Confidence:** 3

**Summary:**

This work identifies limitations in existing machine unlearning evaluation metrics, demonstrating their inadequacy in at least one of three properties: practicality, exactness, and robustness. The authors then propose a new metric called Distribution Correction-based Unlearning Evaluation (DCUE). DCUE operates by identifying core tokens from the dataset, correcting the distributional biases between the original model ($M_o$) and the ground-truth unlearned model ($M_r$), and quantifying the unlearning effectiveness using a Kolmogorov-Smirnov (KS) test. By applying DCUE to evaluate existing unlearning methods, the work also shows that current mechanisms do not truly achieve complete machine unlearning.

**Strengths:**

1. This paper identifies limitations in existing machine unlearning evaluation metrics and introduces a novel metric to address them. Furthermore, it evaluates current unlearning algorithms, providing valuable insights for the design of future methods.

**Weaknesses:**

1. The authors' definition of robustness seems to be flawed. Specifically, the authors claim that *If the post-processing operations are independent of $D_f$ , such as PostProft: $M_u$ undergoes fine-tuning on $D_u$, where $D_u \cap D_f = \varnothing$*. This robustness property fails to account for the fact that such fine-tuning on $D_u$ can act as a form of knowledge dilution, systemically reducing a model's memorization of $D_f$ [1]. A change in the evaluation score in this case does not indicate a lack of robustness in the metric, but rather accurately reflects a genuine change in the model's state.

2. The paper asserts the necessity of its core token identification mechanism based on the ablation study in Section 5.2. However, the data in Figures 7 and 8 do not strongly support this claim. The mechanism appears to have a negligible impact in most cases, with scores remaining the same across all properties on the Llama2-7B model. The sole exception is a drop in the PostProft property for phi-1.5 from 1.0 to 0.83. As argued previously, the validity of PostProft as a property is itself questionable. As this mechanism is one of the paper's key novelties, it requires more robust evidence.

[1] Sun, Chen, et al. "How new data permeates LLM knowledge and how to dilute it." arXiv preprint arXiv:2504.09522 (2025).

**Presentation.** Overall, the paper is well-structured and clearly written. However, the clarity of the figures and some of the writings could be improved:

1. Figure 1 should be revised for better readability. With the writing, we know that Mu2 receives higher Rouge-score, but it was not demonstrated in the diagram.
2. Figure 4 requires a larger font size in its legend for readability, and the typo in "targed" should be corrected to "target."
3. At the end of section 4.1, the definition of validation dataset Dv requires clarification. The text states that “Specifically, the validation dataset is required to satisfy two conditions: (1) not included in the forget dataset Df , and (2) not involved in model fine-tuning training”. These conditions appear redundant. Isn’t condition (1) a subset of condition (2)? The term "model fine-tuning training" is also ambiguous

**Questions:**

1. Regarding Table 3, the paper notes that unlearning methods fall short of "complete unlearning" because their scores are closer to $M_t$ than to $M_r$. Given that these scores remain so much closer to $M_t$, does this suggest that the current methods are largely ineffective, rather than merely "incomplete"?

---

### Note · Authors · 2025-12-02

I have read and agree with the venue's withdrawal policy on behalf of myself and my co-authors.